# Single-cell transcriptomic analysis reveals the association of Ccl6⁺Ccr2⁺Arg1⁺ macrophages with renal interstitial fibrosis in AKI

Xin Zheng[1,2☯†], Jiayu Wang [1,2☯†], Weinan Chen[3,4], Qingquan Xu[3,4], Tao Xu[3,4], Liang Chen [3,4]*

1 Department of Urology, Beijing Chao-Yang Hospital, Capital Medical University, Beijing, China,
2 Institute of Urology, Beijing Chao-Yang Hospital, Capital Medical University, Beijing, China,
3 Department of Urology, Peking University People's Hospital, Beijing, China, 4 Peking University Applied Lithotripsy Institute, Beijing, China

☯ These authors contributed equally to this work.
† Co-first authors.
* chenliang0327@sohu.com

## Abstract

### Background

Acute kidney injury (AKI) is a major health burden with a high risk of progression to chronic kidney disease (CKD). Renal fibrosis is the ultimate outcome of CKD progression, with M2 macrophages playing a critical role by secreting pro-fibrotic factors. Chemokines can influence the progression of renal fibrosis by modulating macrophage polarization during the course of AKI.

### Methods

An integrative analysis of single-cell transcriptomic data from kidneys of mice 7 days after AKI was performed to investigate ligand-receptor (LR) interactions between macrophages and to explore gene co-expression patterns during macrophage differentiation under AKI conditions. The AKI model was induced by unilateral ischemia-reperfusion injury (uIRI), and kidney samples were harvested at day 7. qPCR and WB were employed to measure the transcriptional levels of Ccl6, Ccr2, and M2 polarization markers in macrophages. Transwell assays were performed to evaluate the effect of Ccl6 on BMDMs migration. Cell proliferation was assessed using the Cell Counting Kit-8 (CCK-8) assay. Histological analysis was performed to assess the extent of kidney injury and fibrosis. Multiplex immunofluorescence analysis was conducted to assess the co-localization of Ccl6, Ccr2, and Arg1 expression.

### Results

Through integrated analysis of multiple single-cell transcriptomic datasets from AKI, we identified strong interactions between Ccl6 and Ccr2 in renal macrophages at 7

**Data availability statement:** All relevant data are within the manuscript and its Supporting Information files.

**Funding:** The author(s) received no specific funding for this work.

**Competing interests:** The authors have declared that no competing interests exist.

**Abbreviations:** AKI: Acute kidney injury; CKD: Chronic kidney disease; MMT: Macrophage-to-myofibroblast transition; ESRD: End-stage renal disease; OSOM: Outer stripe of the outer medulla; IRI: Ischemia-reperfusion injury; uIRI: Unilateral ischemia-reperfusion injury; scRNA-seq: Single-cell RNA sequencing; PCA: Principal component analysis; DEGs: Differentially expressed genes; LR: Ligand-receptor; PPI: Protein-Protein Interaction.

days post-AKI. Additionally, co-expression of Ccl6, Ccr2, and Arg1 was observed in renal macrophages, and the abundance of Ccl6⁺Ccr2⁺Arg1⁺ cells was positively correlated with the severity of renal interstitial fibrosis. Ccl6 promoted the migration and M2 polarization of bone marrow-derived macrophages (BMDMs). Inhibition of Ccr2 in AKI mice reduced the infiltration of Arg1⁺ macrophages and attenuated the progression of renal fibrosis.

## Conclusion

Targeting the Ccl6/Ccr2 axis may attenuate fibrotic progression and offers potential therapeutic insights for preventing the transition from AKI to CKD, a possibility that warrants further validation through future functional experiments.

## Introduction

AKI remains associated with high mortality over the past five decades, with severe AKI potentially leading to the development of CKD and eventually progressing to end-stage renal disease (ESRD) [1]. CKD affects 8–15% of the global population, with renal fibrosis being the final outcome of CKD progression, resulting in loss of kidney function in affected patients [2,3]. Recent studies have revealed that persistent autophagy following acute kidney injury (AKI) can induce a pro-fibrotic phenotype in tubular epithelial cells, thereby promoting maladaptive renal repair and exacerbating renal fibrosis [4], Moreover, exosomes derived from tubular epithelial cells have been shown to activate myofibroblasts, further accelerating the progression of renal fibrosis [5].

Injury-induced alterations in the renal microenvironment lead to a dynamic evolution of infiltrating macrophage phenotypes during the repair and tissue remodeling phases [6]. In ischemia-reperfusion injury (IRI), early infiltrating macrophages predominantly exhibit a pro-inflammatory M1 phenotype, whereas during the tissue repair phase, they largely display an M2 phenotype [7]. Macrophages are the primary cell type contributing to renal fibrosis and progressive scarring [8]. Macrophage polarization plays a pivotal role in modulating renal inflammation during AKI, facilitating the repair of injured tubular epithelial cells, and promoting subsequent fibrotic progression [9]. The mechanisms of M2 macrophage polarization involve multiple signaling pathways and molecules, typically induced by IL-4 and IL-13 [10], These cytokines activate the STAT6 signaling pathway, suppress numerous inflammatory enhancers, and promote the expression of M2-related genes such as Arg1, Chi3l3, and Retnla [11–13]. CD206⁺M2 macrophages are closely associated with renal fibrosis in both human and animal models [7]. M2 macrophages can undergo macrophage-to-myofibroblast transition (MMT) and directly differentiate into α-SMA⁺ myofibroblasts [14–16], MMT is mediated by the TGFβ1-Smad3 signaling pathway and regulated by the Src kinase network. Upon binding of TGFβ1 to its receptor, Smad3 activates Src, which in turn promotes myofibroblast proliferation and activation [17,18]. Furthermore, M2 macrophages promote renal fibrosis by secreting

pro-fibrotic factors such as TGFβ, matrix metalloproteinases (MMP-2, MMP-9, MMP-12), galectin-3, and platelet-derived growth factors, which activate fibroblasts and drive extracellular matrix accumulation [7].

Chemokines are secreted proteins with a molecular weight of approximately 8–12 kDa and serve as key mediators that guide the directional migration of immune cells toward sites of tissue injury [19]. By binding to their corresponding chemokine receptors, they play crucial roles in regulating inflammatory responses, maintaining immune homeostasis, and influencing tumor progression [20]. Moreover, chemokines are involved in the pathogenesis and progression of various diseases, including hepatitis [21], atrial fibrillation [22], and fibrosis [23]. Chemokines exert immunomodulatory effects by influencing macrophage polarization. For example, Ccl3 promotes M1 macrophage polarization, thereby inhibiting tumor progression and enhancing chemosensitivity [24]. Inhibition of Ccl2 in macrophages leads to upregulation of M1-associated genes and a concomitant reduction in M2 marker expression [25]. Under inflammatory conditions, CXCL10 interacts with CXCR3 to promote M1 polarization of macrophages, whereas in non-inflammatory contexts, it favors M2 polarization [26].

In kidney injury models, Ccl6 acts as a chemokine that promotes M2 polarization through activation of the PI3-kinase/Akt signaling pathway [27], Inhibition of Ccr2 on the surface of macrophages reduces their infiltration into peripheral tissues [28,29], Following kidney or myocardial injury, the early infiltration of Ccr2$^+$ macrophages is closely associated with tissue fibrosis [29,30]. Multiple studies have demonstrated that monocyte *Ccr2*-deficient mice exhibit reduced fibrosis across various injury models, including those affecting the heart, kidney, and liver [31–33].

Arg1 has been used as a marker for M2/pro-fibrotic macrophage subpopulations in multiple studies [34–37]. For instance, Arg1 and Fn1 serve as markers of pro-fibrotic macrophages in murine myocardial infarction, where macrophages promote fibroblast activation by producing Arg1 and Fn1 [38]. In renal ischemia-reperfusion injury (IRI), Arg1 expression is upregulated in Ccr2$^+$ macrophages, and Arg1$^+$Ccr2$^+$ macrophages infiltrate the peri-medullary region, the most severely damaged area, facilitating renal tissue regeneration and repair [39]. This suggests that Arg1$^+$Ccr2$^+$ macrophages play a crucial role in the repair process following tubular injury. However, it remains unclear whether Ccl6 is co-expressed with Arg1 and Ccr2 in macrophages and whether their expression is correlated with the degree of fibrosis post-AKI.

Our study, through an integrated analysis of single-cell transcriptomic data at day 7 post-AKI, revealed a significant upregulation of *Ccl6*, *Ccr2* and *Arg1* transcription levels in macrophages at this time point. We also confirmed that Ccl6$^+$Ccr2$^+$Arg1$^+$ macrophages exhibit increased infiltration at day 7 post-AKI, with their infiltration level positively correlated with the degree of renal fibrosis. Inhibition of Ccr2 in AKI mice resulted in reduced infiltration of Arg1$^+$F4/80$^+$ cells. In vitro experiments further confirmed that Ccl6 promotes the migration of bone marrow-derived macrophages (BMDMs) and upregulates the expression of the M2 polarization marker Arg1. Targeting the Ccl6/Ccr2 axis may help delay the progression of renal fibrosis post-AKI and provide novel theoretical insights and potential therapeutic targets for mitigating the transition from AKI to CKD and developing anti-fibrotic therapies, pending validation in future functional studies.

## Materials and methods

### Mice

Male C57BL/6 mice (8–10 weeks old, 20–25g) were obtained from the Vital River
(Beijing Vital River Laboratory Animal Technology Co., Ltd.). All experiments were approved by the Animal Ethics Committee of Capital Medical University and conducted in strict accordance with the " China Guide for the Protection and Use of Laboratory Animals ".

### Establishment of AKI model

In this study, male C57BL/6 mice were randomly divided into three groups (n = 15): One group underwent unilateral ischemia-reperfusion injury (uIRI) to induce acute kidney injury (AKI) (n = 6), while another group served as the

sham-operated control (n = 6). In addition, three AKI mice received treatment with the INCB3344 and were designated as the Ccr2 antagonist treatment group (n = 3). Mice in the uIRI group underwent retroperitoneal clamping of the renal pedicle, with the surgery performed in a temperature-controlled room (25°C). Anesthesia was administered using ketamine (80–100 mg/kg, intraperitoneal) and xylazine (10 mg/kg, intraperitoneal). The left renal pedicle was clamped for 30 minutes to ensure ischemia. The sham group underwent the same procedure without renal pedicle clamping. In the Ccr2 antagonist treatment group, AKI mice received intraperitoneal injections of INCB3344 (Abcam, HY-50674) at 30 mg/kg starting one day before surgery and continued once daily postoperatively. Kidney and peripheral blood samples were collected for analysis 7 days after uIRI-induced AKI. Mice were euthanized by cervical dislocation under deep anesthesia induced by an intraperitoneal injection of ketamine (80–100 mg/kg) and xylazine (10 mg/kg), in compliance with the AVMA Guidelines for the Euthanasia of Animals (2020) and approved by the Institutional Animal Care and Use Committee (IACUC) of Capital Medical University.

## Data collection from public databases

We carefully selected single-cell RNA sequencing (scRNA-seq) datasets and ensured that the AKI models were consistently constructed. The following datasets were integrated for analysis, all of which used uIRI for 27–30 minutes to induce AKI, with kidney samples collected 7 days post-surgery for single-cell RNA sequencing. The mouse AKI scRNA-seq datasets were sourced from GSM4142627, GSM4142628, and GSM5815631, while the control group datasets were obtained from GSM4142623, GSM5815627, GSM5911966, and GSM5911967.

## Quality control

For downstream analysis, we utilized Seurat (v5.0.0) to process the single-cell expression matrix. Initial quality control was performed to remove low-quality cells based on the following criteria: (1) cells with ≤ 200 or ≥ 2500 detected genes were excluded, (2) cells with ≤ 200 or ≥ 15000 UMIs were filtered out, (3) cells with high complexity, indicated by log10 (GenesPerUMI) ≤ 0.8, were removed, (4) cells containing ≥ 5% mitochondrial genes were discarded, and (5) doublets were identified and excluded using the DoubletFinder (v2.0.3) algorithm. Expression data were normalized using the LogNormalize method, and 1500 highly variable genes were selected via the *FindVariableGenesFeature* function based on mean expression and dispersion. After filtering, a total of 37,218 cells remained for further analysis.

## Batch effect correction

Following quality control, we obtained a refined single-cell dataset. Principal component analysis (PCA) on highly variable genes was performed using the *RunPCA* function, and batch effects were corrected with the Harmony (v1.1.0) R package. The *ElbowPlot* function identified the top 19 harmony embeddings, which were used for clustering and visualization. Clustering was achieved using the *FindClusters* function, with resolutions ranging from 0.1 to 1.0, and optimal resolution was selected based on cluster stability assessed by the clustertree (v0.4.3) package. To maintain global structure, we applied the *RunUMAP* function in Seurat with the first 18 principal components, resulting in 22 clusters at a resolution of 0.5.

## DEGs analysis and Cell Type Identification

The FindAllMarkers function in Seurat was employed to identify differentially expressed genes (DEGs) across all clusters in the scRNA-seq datasets. The 22 clusters were initially annotated using the MouseRNAseqData from the celldex package, and SingleR identified 8 distinct cell types. The annotations were manually refined and calibrated using CellMarker 2.0 (http://117.50.127.228/CellMarker/) [40]

## Cell chat analysis

Cell–cell communication was analyzed using the CellChat package [41] with the CellChatDB.mouse database, which includes ligand-receptor (LR) interactions. The Seurat object was processed in CellChat following standard protocols.

A weighted directed network of significant LR interactions between cell types was constructed by calculating LR numbers and interaction weights. Signal strength for significant signals was computed in each cell type, and outward/inward degrees were used to identify key senders, receivers, mediators, and influencers.

## Pseudobulk analysis

Pseudobulk RNA-seq analysis was conducted using the DESeq2 package [42]. The gene expression matrix for tubular cells was extracted from the Seurat object via the *aggregateExpression* function, using the 'RNA' assay and 'counts' slot to obtain raw counts. These counts were summed by gene, forming a consolidated expression matrix, which was normalized with DESeq2 by estimating size factors and applying the *counts* function. The resulting matrix provided normalized gene counts for each cell, similar to bulk sequencing output. Differential expression analysis was performed using the DESeq method with a negative binomial distribution. Pairwise comparisons between AKI and control groups were made with the Wald test, using an α-value of 0.5 for independent filtering. $Log_2FC$ values were computed and adjusted via the *lfcShrink* function with the ashr method. DEGs were then selected for further analysis based on an adjusted *p*-value $< 0.05$ and $|log_2FC| > 1$.

## GO enrichment analysis

GO analyses were conducted using the clusterProfiler package [43], selecting biological process GO terms with p-value $< 0.05$. The results were visualized using the ggplot2 package.

## Pseudotime analysis

Pseudotime analysis was conducted with Monocle3 [44], using UMAP and gene expression data to build trajectories via learn_graph (minimal_branch_len $= 4$) and order_cells for pseudotime calculation. Root nodes were manually set. DEG-related expression was fitted with smooth.spline and visualized, clustering genes using k-means with ComplexHeatmap [45].

## PPI network analysis

Protein-Protein Interaction (PPI) network analysis was performed using Metascape (https://metascape.org/), a comprehensive bioinformatics platform that integrates multiple authoritative databases to identify and visualize interactions between proteins.

## RT-qPCR

All real-time reactions were conducted on a 7500 Real-Time PCR System (Thermo Fisher Scientific) using the following primers: Ccl6: 5'-ACTCCAAGACTGCCATTTCATT-3' (Forward) and 5'-AAGCAGCAGTCTGAAGAAGTGTCT-3' (Reverse); Ccr2: 5'-ACGATGATGGTGAGCCTTGTC-3' (Forward) and 5'-TGCAGCATAGTGAGCCCAGA-3' (Reverse); Retnla: 5'- CCAATCCAGCTAACTATCCCTCC-3' (Forward) and 5'- CCAGTCAACGAGTAAGCACAG-3' (Reverse); Arg1: 5'-CTCCAAGCCAAAGTCCTTAGAG-3' (Forward) and 5'- AGGAGCTGTCATTAGGGACA-3' (Reverse). GAPDH: 5-AACTTTGGCATTGTGGAAGGGCTC-3 (Forward) and 5-TGGAAGAGTGGGAGTTGCTGTTGA-3 (Reverse).

## Western blot

Protein concentrations were measured with the BCA Protein Assay Kit (Thermo Fisher Scientific). Equal protein amounts were separated by SDS-PAGE. After blocking with 5% non-fat milk, membranes were incubated with primary antibodies overnight at 4°C, followed by fluorescent secondary antibodies. The primary antibody used was Ccr2a (Proteintech Group, Inc, 16153–1-AP, using at a 1:1000 dilution), Signals were captured using an Invitrogen iBright imager (Thermo Fisher Scientific), GAPDH (Proteintech Group, Inc., 60004−1- Ig, using at a 1:20000 dilution).

## Isolation of Bone Marrow-Derived Macrophages (BMDMs)

Under anesthesia, mice were humanely euthanized via cervical dislocation and disinfected in 75% ethanol for 10 minutes. All procedures were conducted aseptically in a biosafety cabinet. Both femurs and tibias were isolated, and the bone ends were trimmed. Bone marrow was extracted by flushing with RPMI-1640 medium using a 10 mL sterile syringe, repeated four times. The suspension was passed through a 200-mesh strainer and centrifuged at 1200 rpm for 5 minutes. After removing the supernatant, cells were treated with ~5 mL of sterile $1 \times$ red blood cell lysis buffer ($10 \times$ cell volume), gently resuspended, and lysed on ice for 5 minutes. Following centrifugation at 1000 rpm for 5 minutes, the red supernatant was discarded, and the pellet was washed twice with RPMI-1640 medium. The isolated BMDMs were utilized in downstream experiments.

## CCK-8 assay

BMDM cells were plated at a density of $7 \times 10^4$ cells per well in 700 μL of BMDM-specific medium (FuHeng Bio, PY-M079) in a 24-well plate. Cells were then treated with 100 ng/mL Ccl6 (Abcam, HY-P7143) or an equivalent volume of vehicle for 24 hours. After treatment, 70 μL of CCK-8 solution (Lablead, China, CK001) was added to each well, and the plate was incubated for 2 hours in a $CO_2$ incubator. Absorbance at 450 nm was measured using a microplate reader (Varioskan Flash 3001, Thermo Fisher Scientific).

## Transwell assay

In the lower chamber of the Transwell system, 700 μL of BMDM-specific medium was added. The experimental group was treated with 100 ng/mL Ccl6 (Abcam, HY-P7143), while the control group received an equal volume of vehicle. The BMDMs were then transferred to BMDM-specific medium ($5 \times 10^5$ cells/well, 200 μL) and plated in the upper chamber of a Transwell system. The system was incubated in a $CO_2$ incubator for 24 hours. The Transwell upper chamber (Corning) featured a polycarbonate membrane with an 8 μm pore size. Following incubation, the cells were subjected to crystal violet (Leagene, DZ0062) staining to assess migration. Images were captured using Leica DMi1 inverted microscope (Leica Camera AG, German), and the number of migrated BMDMs was quantified using Fiji (an image processing software based on ImageJ).

## Flow cytometry

BMDMs were treated with 100 ng/mL Ccl6 (Abcam, HY-P7143) or an equal volume of vehicle for 24 hours. After stimulation, BMDMs were collected and prepared as a single-cell suspension. The BMDMs were then stained with Fixable Viability Stain 700 (BD Horizon™, 564997) and Mouse BD Fc Block (BD Pharmingen™, 553141) to exclude dead cells and block non-specific binding. For intracellular staining, cells were permeabilized and incubated with Arg1 antibody (Bioss, bs-8585R-BF488) at 4°C for 45 minutes. After washing with PBS, flow cytometry was performed using the BD LSRFortessa™ (BD, USA), and data were analyzed with FlowJo V10.

## Immunofluorescence

Fluorescent immunohistochemistry with tyramide signal amplification was performed on 4 μm TMA sections. Slides were deparaffinized (xylene, graded ethanol, and 10% neutral buffered formalin for 10 minutes) and washed three times with distilled water and Tris-buffered saline/Tween, each for 5 minutes. Antigen retrieval was achieved by microwave heating in citrate buffer (full power for 45 seconds, then 20% power for 15 minutes), followed by washing. Tissue blocking was done with a protein blocker (X9090, Dako), and primary antibodies were applied for 30 minutes before washing. Slides were then incubated with HRP-conjugated secondary antibodies for 10 minutes. Tyramide signal amplification (Perkin Elmer) was applied for 10 minutes, followed by washing. Before staining additional markers, antigen retrieval and blocking steps

were repeated. Single-stained slides were included in each cycle as controls. The primary antibodies used were Ccl6 (Bioss, Beijing Biosynthesis Biotechnology Co., Ltd. bs-2476R, using at a 1:100 dilution), Ccr2 (Proteintech Group, Inc., 16153–1-AP, using at a 1:100 dilution), and Arg1 (Proteintech Group, Inc., 66129–1-Ig, using at a 1:2000 dilution). F4/80 (Proteintech Group, Inc., 29414–1-AP, using at a 1:200 dilution).

### H&E staining

Kidney tissues were preserved in 4% paraformaldehyde, processed for paraffin embedding, and sectioned for H&E staining [46,47]. Tubular injury was assessed based on H&E staining results as previously described [48]. The evaluation included tubular necrosis, cast formation, and brush border loss, and was independently performed by two blinded investigators. For scoring, three random high-power fields were analyzed, and the percentage of affected tubules was determined using the following scale: 0 (no injury), 1 (1%−25%), 2 (26%−50%), 3 (51%−75%), and 4 (75%−100%).

### Picrosirius red staining

Kidney tissue sections (4–6 μm) were deparaffinized in xylene and rehydrated through a graded ethanol series to distilled water. The sections were stained with Sirius Red solution (0.1% Sirius Red in saturated aqueous picric acid) for 1 hour at room temperature to label collagen fibers. After staining, the sections were rinsed in 0.5% acetic acid to remove excess dye and washed briefly in distilled water. The sections were then dehydrated using graded ethanol, cleared in xylene, and mounted with neutral balsam. Under polarized light microscopy, collagen fibers appeared as bright red or orange, while non-collagenous structures were unstained or faintly yellow.

### Masson's trichrome staining

Kidney tissue sections (4–6 μm) were deparaffinized in xylene and rehydrated through a graded ethanol series to distilled water. Sections were stained with iron hematoxylin for 8 minutes to visualize nuclei, followed by thorough washing with running water. Acid fuchsin was then applied for 3 minutes to stain cytoplasm and muscle fibers, and sections were subsequently treated with phosphotungstic acid or phosphomolybdic acid for 10 minutes to remove nonspecific staining and enhance collagen fiber staining specificity. Collagen fibers were stained using aniline blue for 5 minutes, followed by washing with running water. The sections were then dehydrated in graded ethanol, cleared in xylene, and mounted with neutral balsam. The staining results showed nuclei and glomeruli as deep blue or black, cytoplasm and muscle fibers as red, and collagen fibers (e.g., in the basement membrane and fibrotic areas) as blue.

### Statistical analysis

Single-cell transcriptomic data were analyzed in the R (4.2.2) environment, with all scripts executed in RStudio. To assess differences between groups, a two-tailed Student's t-test was applied, with significance set at $P < 0.05$ to ensure statistical rigor. Correlation analysis was performed using Spearman's coefficient. Data visualization and final statistical analyses were performed using GraphPad Prism 10.

## Results

### The cellular landscape of kidney sample

S1 Fig 1 provides an overview of the flowchart of our study. We integrated the full-kidney scRNA-seq data of uIRI-induced AKI and Sham control mice from the GEO database. To ensure consistent experimental conditions, we performed selection and ultimately chose 3 datasets for the AKI group and 4 datasets for the Control group. After quality control, filtering, and batch effect correction, 37,218 single-cell transcriptomes were obtained. We employed unsupervised clustering (uniform manifold approximation and projection [UMAP]) to generate a single-cell transcriptome atlas of mice kidneys. We

identified 22 distinct cell clusters and annotated them based on expression of known marker genes. Totally, we identified 11 different cell subtypes within these clusters (Fig 1a). We have displayed the marker genes for all subtypes, Proximal tubular cells ($Cml1^{hi}$, $Gsta^{hi}$), Endothelial cells ($Adgrl4^{hi}$, Plpp1$^{hi}$), Epithelial cells ($Slc12a1^{hi}$, $Umod^{hi}$), Adipocyte ($Hmx2^{hi}$, $Aqp^{hi}$), Fibroblasts ($Plac9a^{hi}$, $Plac9b^{hi}$), Macrophages ($Lyz1^{hi}$, $Ms4a7^{hi}$), T cells ($Cd3d^{hi}$, $Cd3e^{hi}$), B cells ($Cd19^{hi}$, $Cd79a^{hi}$), NK cells ($Klra4^{hi}$, $Ncr1^{hi}$), Monocytes ($Cd209a^{hi}$, $Ifitm6^{hi}$), Neutrophils ($S100a8^{hi}$, $S100a9^{hi}$) (Fig 1b). Additionally, we generated cell atlases based on different groups (AKI vs Control) (Fig 1a). To assess the differences in cell type abundance between the AKI and Control groups, we compared the number and proportion of cell clusters between the two groups (Fig 1c, 1d). It can be observed that compared to the Control group, AKI resulted in a decrease in the number and proportion of proximal tubular cells and renal tubular epithelial cells, while the number and proportion of immune cells significantly increased, indicating inflammatory cell infiltration. Among the AKI group, the proportion of macrophages showed the most significant increase (26.7%) (Fig 1c), highlighting the important role of macrophages in AKI. Furthermore, we compared the within-group cell composition differences between different AKI datasets (AKI.1, AKI.2, AKI.3) and the Control group. It was found that the proportion of macrophages in all AKI datasets was significantly higher than that in the Control group (Fig 1d).

## Cell-cell communication revealed that migration was triggered among macrophages via the Ccl6-Ccr2 axis

Given the significant changes in macrophage abundance in the AKI group, likely resulting from the complex intercellular communication induced by uIRI, we employed CellChat to explore the interaction network between macrophages and other cells. The results revealed differences in the number and strength of interactions among cell subtypes in both the

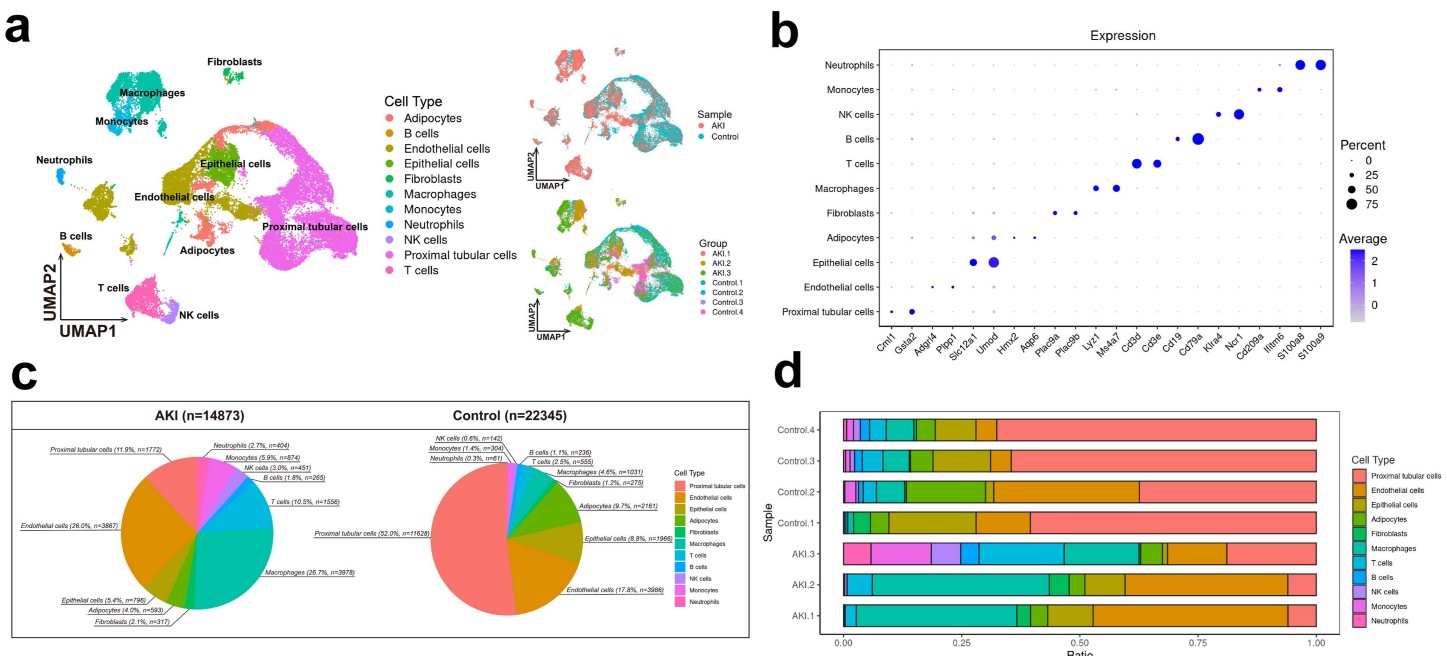

**Fig 1. The cellular landscape of the kidney 7 days after uIRI-induced AKI and in Sham control mice.** (a) Single-cell transcriptome atlas of mouse kidneys identifies 11 cell types, including proximal tubular cells, endothelial cells, epithelial cells, adipocytes, Fibroblasts and six types of immune cells (monocytes, neutrophils, T cells, B cells, NK cells, and macrophages). (b) Marker gene expression for each cell type, with dot size indicating the proportion of expressing cells and color representing average expression level. (c) Pie charts showing decreased proximal tubular and epithelial cells and increased immune cells, particularly macrophages (26.7%), in AKI kidneys compared to controls. (d) Bar plots of cell type proportions across datasets confirm increased macrophage infiltration in all AKI samples relative to controls.

AKI and sham control groups (Fig 2a). Notably, we observed significant alterations in interactions between macrophage subtypes and other cells, with markedly increased interaction number and weight between macrophages and themselves, as well as with neutrophils and lymphocytes in the AKI group (Fig 2a). Subsequent analysis of the ligand-receptor interaction weights demonstrated that macrophages in the AKI group exhibited enhanced signal reception and outgoing (Fig 2b). Furthermore, a comparison of intercellular signaling pathways between the AKI and Control groups revealed significantly intensified interactions among immune cell subtypes in the AKI group, particularly within macrophage subtypes (Fig 2c). By examining the differences in ligands and receptors involved in macrophage signaling pathways between the two groups, we found that the CCL signaling pathway was significantly upregulated in macrophages in the AKI group, suggesting activation of this pathway during AKI (Fig 2d). Further analysis identified the Ccl6/Ccr2 ligand-receptor pair as a key driver of this enhanced signaling (Fig 2e). These findings indicate that at day 7 post-AKI, interactions between macrophage subtypes via Ccl6/Ccr2 may promote the massive migration of bone marrow-derived macrophages (BMDMs) to the injured kidney, potentially influencing the repair processes in injured renal tissue. To validate the robustness of our findings, we conducted individual CellChat analyses on each AKI dataset. We consistently observed activation of the CCL signaling pathway compared to controls. The Ccl6–Ccr2 demonstrated strong interactions and emerged as a significantly enriched communication axis across all AKI datasets at 7 days post-injury (Fig 2f).

### Trajectory analysis revealed temporal expression consistency of *Ccl6*/*Ccr2* in macrophages

To further investigate the key macrophage subpopulations in AKI kidneys, we selected 5,009 macrophages from the AKI and Control groups. After correcting for batch effects, we performed dimensionality reduction and reclustering, resulting in a refined macrophage subpopulation atlas with six distinct clusters. Notably, these macrophage subpopulations displayed a temporal pattern: Cluster 1 and Cluster 2 (Stage I) were derived from both Control and AKI mouse kidney, whereas Cluster 0, Cluster 3 and Cluster 4 (Stage II) predominantly originated from AKI mouse kidney tissues (Fig 3a). A heatmap highlighted the marker genes for these macrophage subpopulations (Fig 3b).

Cluster 0 consists of pro-repair macrophages, primarily expressing *Cp* and *Sepp1*, which are involved in anti-inflammatory responses, antioxidation, and metal homeostasis regulation. Cluster 1 is characterized by high expression of *Rpl24* and *Rpl30*, both of which are ribosomal proteins responsible for regulating protein synthesis, inflammation modulation, and cellular metabolism in macrophages. Cluster 2 represents pro-inflammatory macrophages, marked by the expression of pro-inflammatory genes *Tnf* and *Ccl3*. Cluster 3 consists of macrophages involved in extracellular matrix remodeling, expressing *Spp1*, a key extracellular matrix component, as well as *Trem2*, which plays a role in clearing cellular debris and suppressing macrophage-mediated inflammation. Cluster 4 comprises proliferating macrophages, predominantly expressing cell cycle-related genes such as *Top2a* and *Mki67*. Cluster 5 is likely composed of doublets, as it exhibits high expression of T cell markers *Trbc2* and *Cd3e* (Fig 3c).

To elucidate the differentiation trajectory of macrophage subpopulations 7 days after the onset of AKI, we performed pseudotime analysis using Monocle 3, which revealed a differentiation process from Stage I to Stage II (Fig 3d). The heatmap further displayed the expression of genes along the pseudotime trajectory, with *Ccl6* and *Ccr2* showing significantly elevated expression at the terminal stage of macrophage differentiation. This suggests that the interaction between Ccl6 and Ccr2 likely occurs predominantly at the later stages of differentiation, potentially concentrating within M2 macrophages (Fig 3e). Additionally, FeaturePlot demonstrated that both Ccl6 and Ccr2 were highly expressed in the macrophage subpopulations of Stage II and revealed distinct expression patterns across macrophage clusters. In Cluster 0, the majority of macrophages exhibited high Ccr2 and low Ccl6 expression, suggesting that these cells are likely responding macrophages, such as recruited or activated cells. Notably, a small subset within Cluster 0 co-expressed both Ccl6 and Ccr2 at high levels, indicative of a potential autocrine signaling loop. In contrast, Cluster 3 predominantly expressed Ccl6 alone, supporting a paracrine mechanism whereby Ccl6+ macrophages may serve as a source of ligand to recruit Ccr2+ cells (Fig 3f).

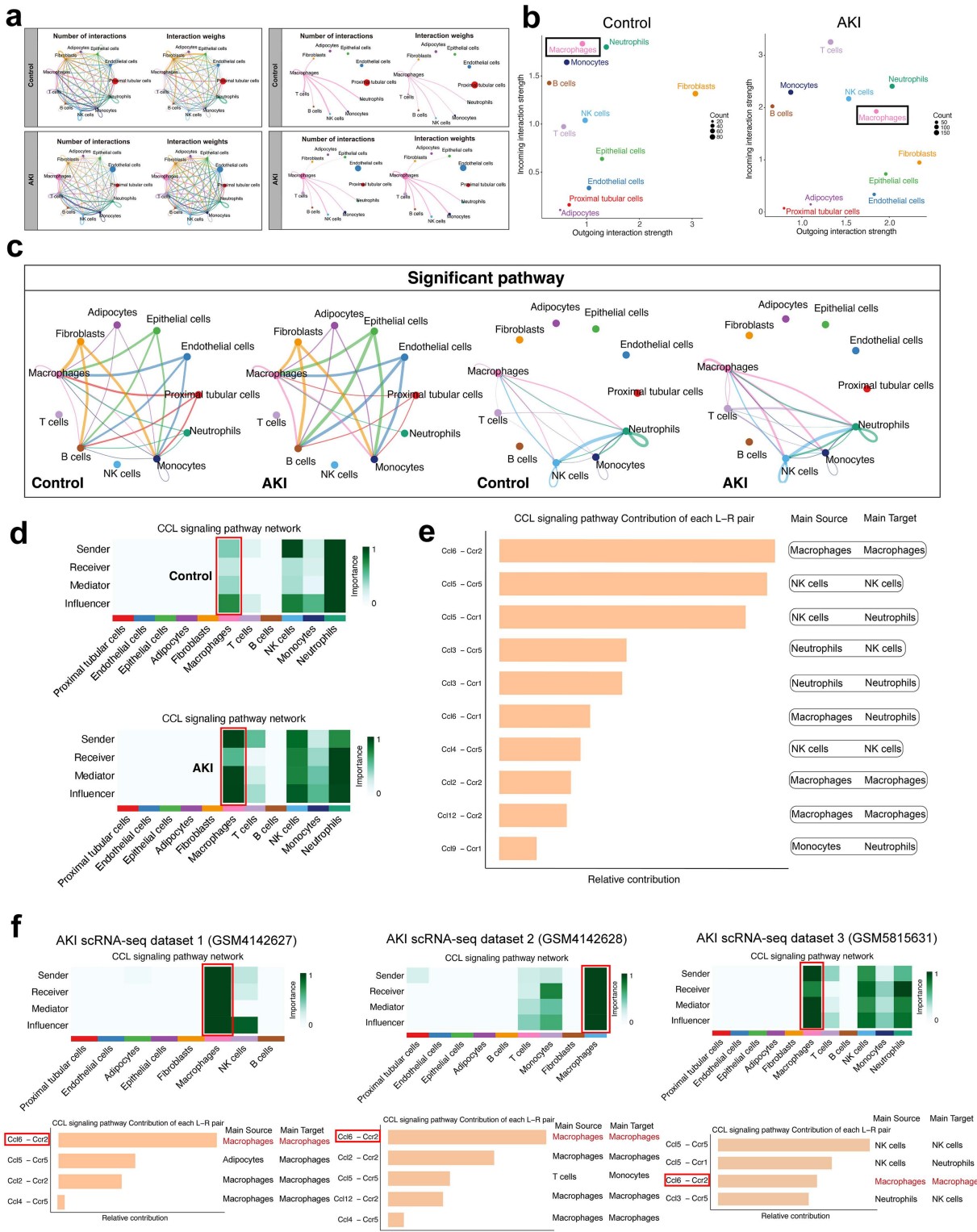

**Fig 2. Ccl6/Ccr2 axis mediates macrophage communication in day 7 post-AKI. (a)** Analysis of cell-cell communication shows differences in interaction number and weight among cell subtypes in AKI and sham control groups. In the AKI group, macrophages exhibit increased interactions with themselves, neutrophils, and lymphocytes. **(b)** Ligand-receptor interaction weights indicate enhanced signal reception and outgoing signaling by

macrophages in AKI kidneys compared to controls. **(c)** Intercellular signaling pathway analysis reveals intensified communication among immune cells, particularly within macrophage subtypes, in the AKI group. **(d)** Chemokine CCL signaling pathways are significantly upregulated in AKI macrophages, indicating activation of this pathway. **(e)** Ccl6/Ccr2 as the key driver of macrophage-macrophage interactions, promoting the recruitment of bone marrow-derived macrophages to the injured kidney and potentially influencing renal repair. **(f)** Validation of CCL signaling pathway activation in AKI. CellChat analyses across all AKI datasets at 7 days post-injury revealed consistent activation of the CCL signaling pathway, with the Ccl6–Ccr2 demonstrating strong interactions and emerging as a significantly enriched communication axis.

### The Ccl6-Ccr2 interaction promotes macrophage migration and M2 polarization

We analyzed the differential gene expression in macrophage subgroups between the AKI group and the Control group. The results showed that compared to the Control group, the AKI group had upregulated expression of 132 genes, including (*Ccl6* and *Ccr2*), and downregulated expression of 295 genes. We displayed the top 5 upregulated and downregulated genes, along with *Ccl6* and *Ccr2*.

We found that M2 macrophage polarization genes such as *Arg1* [49–51] and *Retnla* [52,53], as well as the extracellular matrix (ECM) component *Fn1* [54,55], were significantly upregulated in the AKI group. This indicates that at day 7 after AKI onset, M2 macrophages play a crucial role in renal tissue remodeling (Fig 4a).

The *Ccl6*, *Ccr2* and top 10 differentially expressed genes in macrophage subclusters are predominantly concentrated in Stage II. *Fn1*, *Retnla* and *Arg1* serve as markers for macrophage M2 polarization, confirming the occurrence of M2 polarization in AKI, and suggesting upregulated genes potential involvement in the polarization process (Fig 4b) (Fig 3f). Subsequently, we compared the differences in *Ccl6* and *Ccr2* expression among cell clusters in the AKI group. The results showed that *Ccl6* was mainly highly expressed in Macrophages, Monocytes, and Neutrophils, while *Ccr2* was predominantly expressed in Macrophages, Monocytes, NK cells, and T cells. Notably, *Ccl6* and *Ccr2* exhibited the most significant expression in macrophages (Fig 4c). Correlation analysis revealed a positive correlation between *Ccl6* and *Retnla*, *Ccr2*, *Fn1*, *Gda*, *Arg1*, and *Gpnmb*, while *Ccr2* exhibited a positive correlation with *Retnla*, This indicates that the expression of Ccl6 and Ccr2 is associated with the polarization of M2 macrophages (Fig 4d). In the AKI group, a total of 132 upregulated genes underwent Gene Ontology (GO) enrichment analysis. The results indicated that the upregulated genes were primarily enriched in terms such as cytokine-mediated signal pathway, leukocyte migration, positive regulation in response to external stimuli, myeloid leukocyte migration, and leukocyte chemotaxis (Fig 4e). The heatmap displayed co-expressed genes in the enriched pathways, Ccl6 and Ccr2 are co-enriched across multiple signaling pathways (Fig 4f). According to the gene expression relationship network, it was evident that *Ccl6* and *Ccr2* exhibited co-expression in pathways related to cytokine and chemokine-mediated signal pathways, myeloid leukocyte and leukocyte migration (Fig 4g). Finally, we constructed a PPI network using the upregulated genes from the AKI group. We identified three core MCODE networks. Notably, Ccl6 and Ccr2 showed strong interactions within three pathways: Chemokine receptors binding chemokines, chemokine-mediated signal pathways, and cellular responses to chemokines (Fig 4h). We further characterized *Ccl6⁺Ccr2⁺Arg1⁺* macrophages by examining the expression of additional pro-fibrotic genes, including *Spp1*, *Lgals3*, and *Tgfb1*, using FeaturePlot visualization. We found that *Spp1* and *Lgals3* were highly expressed in the *Ccl6⁺Ccr2⁺Arg1⁺* macrophages, while *Tgfb1* showed low expression levels. To further explore the matrix-remodeling potential of this subset, we analyzed five representative remodeling enzymes: *Mmp9*, *Mmp12*, *Timp1*, *Ctsk*, and *Adamts4*. Among these, only *Mmp12* was highly expressed in the *Ccl6⁺Ccr2⁺Arg1⁺* population, suggesting that these macrophages may contribute to the progression of renal fibrosis through tissue remodeling mechanisms (Fig 4i).

### The infiltration of Ccl6+Ccr2+Arg1+ cells is correlated with the progression of renal interstitial fibrosis after AKI

To investigate findings from single-cell transcriptomic analysis, we utilized the uIRI model to induce AKI and harvested kidney samples at day 7 post-injury (Fig 5a). At this time point, the transcriptional levels of *Ccl6*, *Ccr2*, *Retnla*, and *Arg1* were significantly upregulated, corroborating results from the single-cell dataset (Fig 5b). Histological analysis revealed

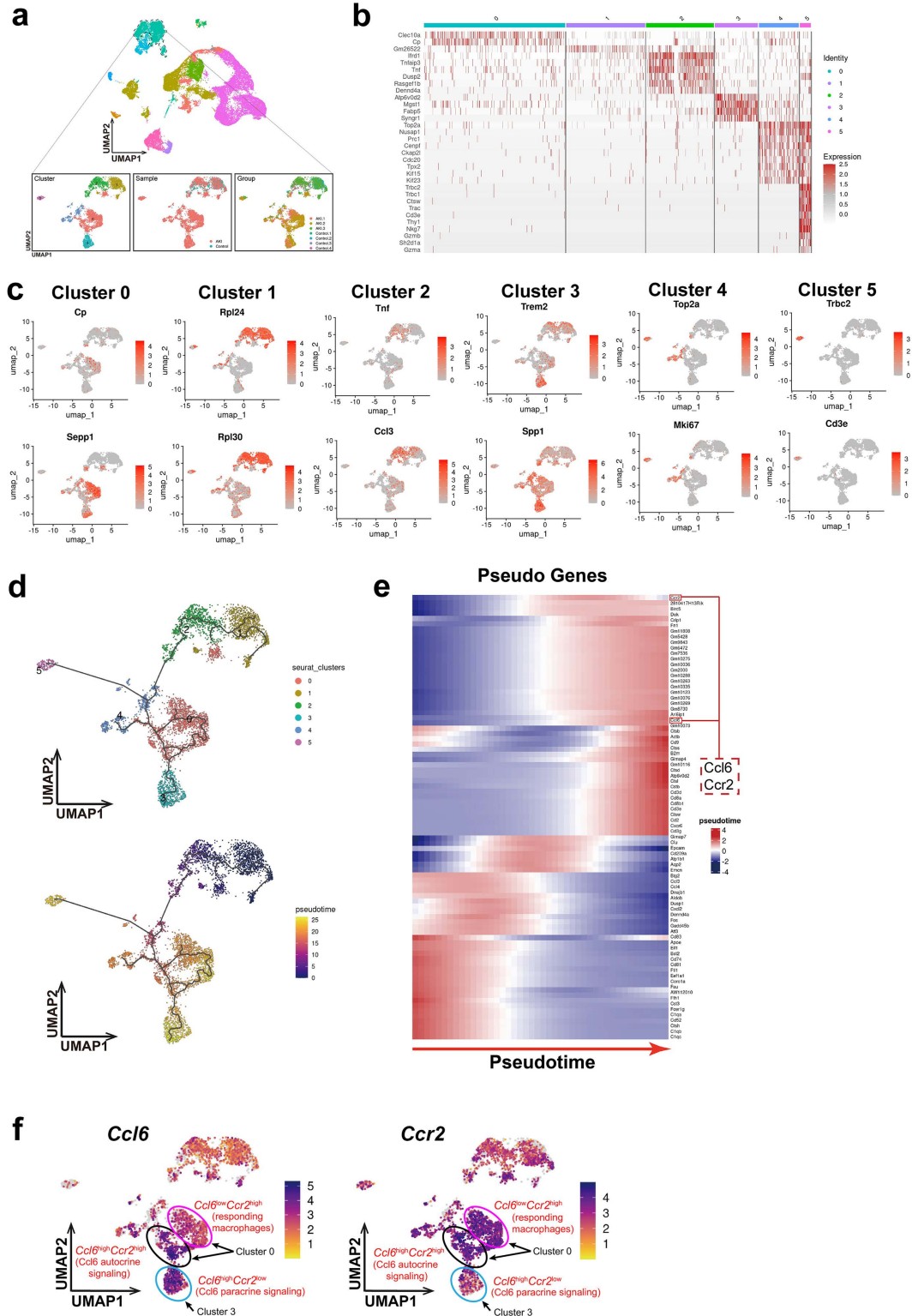

**Fig 3. Trajectory analysis reveals temporal expression consistency of Ccl6 and Ccr2 in macrophages during AKI. (a)** UMAP visualization of 5,009 macrophages from Control and AKI mouse kidney tissues, reclustered into six distinct subpopulations. **(b-c)** Heatmap and FeaturePlot showing the marker genes for each macrophage subpopulation. **(d)** Pseudotime analysis using Monocle 3 reveals a differentiation trajectory from Stage I to

Stage II. **(e)** Heatmap of pseudotime-ordered gene expression, highlighting differentially expressed genes along the trajectory. *Ccl6* and *Ccr2* showed significant expression at the terminal stage of the trajectory. **(f)** FeaturePlot showing the expression patterns of Ccl6 and Ccr2 across macrophage clusters. A subset of Cluster 0 co-expresses both genes, suggesting a potential autocrine loop, while Cluster 3 predominantly expresses Ccl6 alone, supporting a paracrine model.

incomplete renal repair at day 7 post-AKI. H&E staining demonstrated ischemic pathological alterations in the outer stripe of the outer medulla (OSOM) region. Previous studies have shown that the OSOM is the most severely affected region of renal tubular cell injury during AKI [39], including tubular epithelial cell necrosis and detachment, luminal dilation, and brush border loss, with significantly higher tubular injury scores compared to sham-operated controls. Additionally, PSR and Masson staining confirmed the presence of interstitial fibrosis, suggesting that unresolved tubular injury at this stage may be associated with fibrotic repair (Fig 5c; S2 Fig 2a, b, c).

To further examine the expression patterns of Ccl6, Ccr2, and the M2 polarization marker (Arg1) in the kidney at day 7 post-AKI, we performed multiplex immunofluorescence staining. Based on immunofluorescence results, we selected the OSOM region for colocalization analysis. The results showed that the Pearson's correlation coefficients among Ccl6, Ccr2, and Arg1 were significantly increased in the kidneys of mice at day 7 after AKI. Additionally, the Manders' overlap coefficients among the three markers were also elevated (Fig 5d, 5e; S3 Fig 3a, b, c, d, e, f). This indicates that Ccl6, Ccr2, and Arg1 exhibit enhanced co-expression within the same cells in the OSOM region of the kidney at day 7 after AKI. Interestingly, some Ccl6+Ccr2+ cells also co-expressed Arg1, and the number of Ccl6+Ccr2+Arg1+ cells was significantly higher in AKI kidneys compared to sham controls (Fig 5d, 5f). This is consistent with previous findings, as Arg1+ macrophages predominantly infiltrate the OSOM region. These Arg1+ macrophages are involved in the renal repair process following AKI [39]. We found a positive correlation between the number of infiltrating Ccl6+Ccr2+Arg1+ macrophages and the degree of renal interstitial fibrosis in mice following AKI (Fig 5g, 5h).

Western blot analysis of Ccr2 protein levels revealed significant upregulation in AKI kidneys compared to sham controls (Fig 5i; S4 Fig 4a, b). In summary, our study reveals that, seven days after AKI, the transcriptional levels of Ccl6, Ccr2, and Arg1 are upregulated, which is consistent with the results of single-cell RNA analysis. Moreover, Ccl6 and Ccr2 are co-expressed in Arg1+ macrophages. The infiltration of Ccl6+Ccr2+Arg1+ macrophages may promote the progression of renal interstitial fibrosis in AKI.

### The Ccl6/Ccr2 axis regulates macrophage migration, M2 polarization, and renal fibrosis following AKI

To further investigate the biological function mediated by the interaction between Ccl6 and Ccr2, we conducted a series of functional experiments. We re-established the mouse model of AKI, and intraperitoneally administered the Ccr2 antagonist (INCB3344) during the perioperative period (starting one day before surgery and continued once daily postoperatively). On day 7 post-surgery, kidney tissues were harvested to evaluate macrophage infiltration and polarization by co-immunofluorescent staining for Arg1 and F4/80. The results showed that compared with the Sham group, Arg1 and F4/80 expression in AKI kidneys was significantly elevated on day 7. However, after Ccr2 inhibition, their expression was markedly reduced (Fig 6a). Co-localization analysis further revealed significant correlation between Arg1 and F4/80 signals across all three groups, although there were no significant differences in Pearson's correlation coefficient or Manders' overlap coefficient among the groups (Fig 6b, 6c). Next, we quantified the mean fluorescence intensity of Arg1 and the number of Arg1+F4/80+ double-positive macrophages. The results demonstrated that on day 7 after AKI, both Arg1 fluorescence intensity, Arg1-positive area percentage, and Arg1+F4/80+ cell counts were significantly increased. These values decreased markedly following Ccr2 blockade, approaching levels seen in the Sham group (Fig 6d). To further assess the impact of Ccr2 inhibition on renal fibrosis, Masson and Sirius Red staining were performed. Results showed that Ccr2 antagonism significantly alleviated renal fibrosis on day 7 post-AKI (Fig 6e, 6f; S5 Fig b).

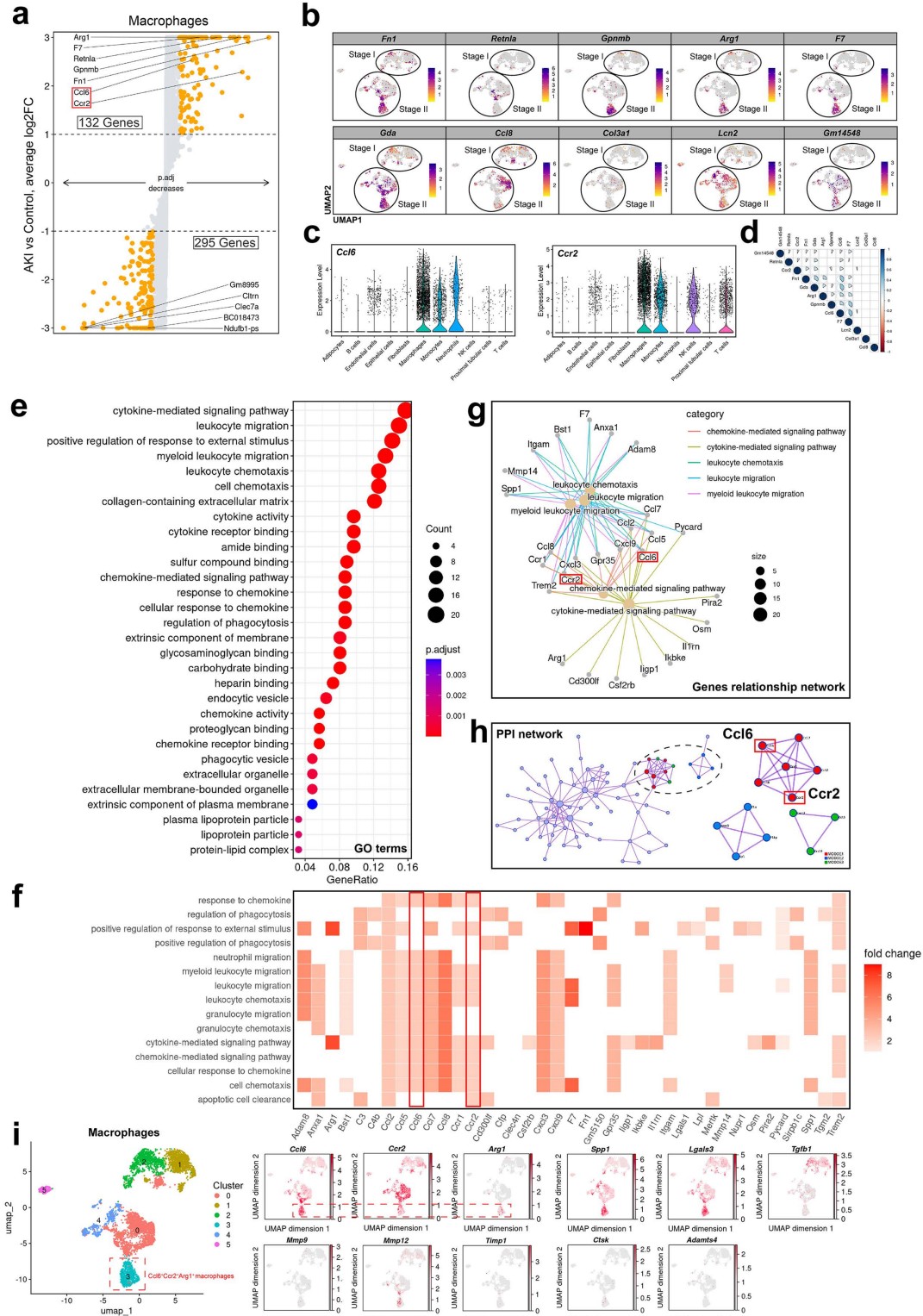

**Fig 4. Ccl6-Ccr2 interaction drives macrophage migration and M2 polarization during AKI. (a)** Differential expression analysis of macrophages in the AKI group compared to the Control group. A total of 132 genes were upregulated, including *Ccl6* and *Ccr2*, while 295 genes were downregulated. **(b)** The FeaturePlot displays the top 10 upregulated genes in macrophages from AKI kidneys. **(c)** UMAP visualization and violin plots showing

the expression distribution of *Ccl6* and *Ccr2* across cell clusters in the AKI group. **(d)** Correlation analysis reveals positive relationships between *Ccl6*, *Ccr2*, and key genes such as *Retnla*, *Fn1*, and *Arg1*. **(e)** GO enrichment analysis of 132 upregulated genes in macrophages from the AKI group. The genes are enriched in pathways such as cytokine-mediated signaling, leukocyte migration, and myeloid leukocyte chemotaxis. **(f)** Heatmap of co-expressed genes involved in the enriched pathways. **(g)** Gene relationship network showing the co-expression of *Ccl6* and *Ccr2* in pathways related to cytokine-mediated signaling, chemokine activity, and leukocyte migration. **(h)** Protein-Protein Interaction (PPI) network of upregulated genes in the AKI group. Ccl6 and Ccr2 demonstrate strong interactions within three core pathways: chemokine receptor-ligand binding, chemokine-mediated signaling, and cellular responses to chemokines. **(i)** *Ccl6*+*Ccr2*+*Arg1*+ macrophages exhibit high *Spp1*, *Lgals3*, and *Mmp12* expression, suggesting pro-fibrotic remodeling potential.

In vitro, we isolated bone marrow–derived macrophages (BMDMs) from mice and confirmed macrophage identity via Cd68 immunofluorescence staining (Fig 6g). CCK8 assays revealed that 24-hour treatment with Ccl6 had no significant effect on BMDM proliferation (Fig 6h). To evaluate the influence of Ccl6 on macrophage migratory capacity, we performed Transwell assays. BMDMs were seeded in the upper chamber, and Ccl6 was added to the lower chamber. After 24 hours, a notable increase in cell migration was observed in the Ccl6-treated group (Fig 6i, 6j; S5 Fig 5c). Finally, flow cytometry was used to assess the polarization of BMDMs after 24-hour Ccl6 treatment. The mean fluorescence intensity of Arg1 was elevated, indicating that Ccl6 promoted macrophage polarization toward the M2 phenotype (Fig 6k, 6l).

Collectively, our in vivo findings demonstrate that pharmacological inhibition of Ccr2 reduces macrophage infiltration and M2 marker (Arg1) expression in the kidney 7 days after AKI, as well as ameliorates renal fibrosis. In vitro results further confirm that exogenous Ccl6 promotes macrophage migration and M2 polarization without affecting proliferation. These findings suggest that the Ccl6/Ccr2 axis may facilitate macrophage recruitment and alternative activation during the later stages of AKI.

## Discussion

Through integrated analysis of scRNA-seq data, our study revealed that seven days post-AKI, macrophages may induce leukocyte migration and chemotaxis through the interaction between Ccl6 and Ccr2, exhibiting a temporal correlation with Arg1 expression. The abundance of Ccl6+Ccr2+Arg1+ cells increase at day 7 post-AKI and is positively associated with the extent of renal interstitial fibrosis. Our findings highlight the therapeutic potential of targeting the Ccl6/Ccr2 axis to mitigate the transition from AKI to CKD, but this requires further validation in future functional studies.

Cell-cell communication analysis of single-cell transcriptomics identified the Ccl6/Ccr2 ligand-receptor pair as a critical interaction within macrophages at day 7 post-AKI. Differential gene expression analysis of macrophage subpopulations further demonstrated significantly increased expression of *Ccl6*, *Ccr2*, and the M2 polarization marker *Arg1* in macrophages during the later stages of AKI. Gene ontology enrichment analysis of upregulated genes indicated their involvement in cytokine-mediated signaling, leukocyte migration, and chemotaxis. PPI network analysis also highlighted strong interactions between Ccl6 and Ccr2 within multiple chemokine-mediated pathways. These findings suggest that the interaction between macrophage-derived Ccl6 and Ccr2 may mediate the chemotaxis and migration of various immune cells during the AKI.

Trajectory analysis of macrophage subpopulations revealed differentiation processes, with *Ccl6* and *Ccr2* expression being upregulated during terminal differentiation. This suggests that their expression is time-specific during macrophage differentiation. We also observed that *Arg1* expression was concentrated in AKI kidneys, indicating a potential shift from a pro-inflammatory M1 phenotype to an anti-inflammatory M2 phenotype. This aligns with previous studies showing that macrophages exhibit an M1 phenotype during the early stages of injury, transitioning to an M2 phenotype during tissue repair [56,57].

Two major macrophage populations exist in the kidney: tissue-resident macrophages (TR-AMs, F4/80high) and monocyte-derived macrophages (Mo-AMs, CD11bhigh, Ly6chigh) that migrate from the peripheral blood to the injured tissue. TR-AMs are critical for maintaining renal homeostasis, repair, regeneration, and inflammatory responses under

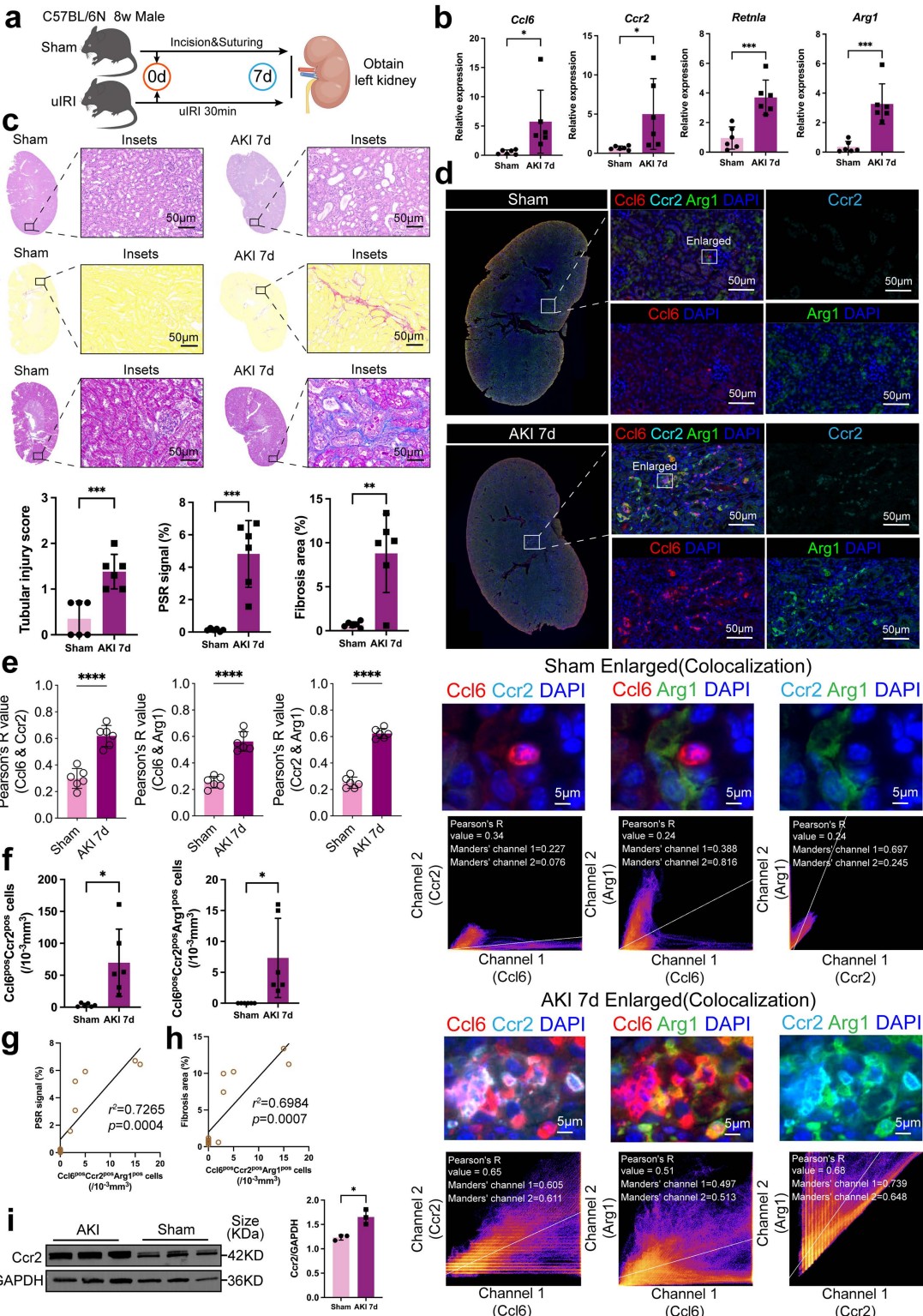

**Fig 5. Ccl6⁺Ccr2⁺Arg1⁺ cell infiltration correlates with the progression of renal interstitial fibrosis following AKI. (a)** Schematic representation of the experimental workflow using the uIRI model to induce AKI. Kidney samples were harvested 7 days post-injury (n = 6). **(b)** Transcriptional analysis of *Ccl6*, *Ccr2*, *Retnla*, and *Arg1* expression levels in kidney tissue, showing significant upregulation in the AKI group compared to sham controls (n = 6).

**(c)** Histological analysis at day 7 post-AKI. H&E staining reveals tubular necrosis, epithelial detachment, and brush border loss. PSR and Masson staining demonstrate significant interstitial fibrosis. Quantification of tubular injury scores and fibrosis areas highlights incomplete renal repair and fibrotic remodeling (n = 6). Scale bar: 50 μm. **(d, e)** Colocalization analysis of Ccl6, Ccr2, and Arg1 in the OSOM region of the kidney at day 7 post-AKI. Immunofluorescence-based quantification showed significantly increased Pearson's correlation coefficients among the three markers, indicating enhanced co-expression within the same cells (n = 6). Scale bar: 50 μm **(f)** Quantification of Ccl6+Ccr2+ and Ccl6+Ccr2+Arg1+ cells revealed a significant increase in kidneys 7 days after AKI compared to sham controls (n = 6). **(g, h)** The infiltration of Ccl6+Ccr2+Arg1+ cells positively correlate with the severity of renal interstitial fibrosis. (i) Western blot analysis of Ccr2 protein levels in kidney tissue shows significant difference between AKI and sham controls. Results are presented as mean ± SD. *P < 0.05, **P < 0.01, ***P < 0.001, *ns*: no significance.

both physiological and pathological conditions through phagocytosis, antigen presentation, inflammation regulation, and extracellular matrix remodeling [58–60]. Mo-AMs, on the other hand, exacerbate inflammation at the injury site and contribute to fibrosis progression, primarily through M2 polarization. Traditionally, M2 macrophages promote type I collagen deposition and the recruitment of fibroblasts expressing α-SMA [7]. M2 macrophages can transition into myofibroblasts (MMT) while secreting IL-1, MMP2, MMP9, and MMP12 to promote the proliferation and activation of myofibroblasts [7,61,62]. Their abundance correlates with tubular injury and fibrosis progression [63], playing a key role in tissue repair and fibrosis following AKI. Thus, inhibiting M2 macrophage polarization may represent an effective anti-fibrotic therapeutic strategy [64].

The interaction between Ccl6 and Ccr2 could occur among TR-AMs, between TR-AMs and Mo-AMs, or among Mo-AMs. Further studies are required to determine the specific macrophage subpopulations involved in this interaction. However, the rapid advancement of single-cell sequencing technology has facilitated the elucidation of macrophage heterogeneity, revealing that the traditional M1/M2 classification is insufficient to capture their complex and diverse biological functions. Multiple studies have demonstrated that Arg1 expression is upregulated following tissue injury, and Arg1+macrophages contribute to tubular cell regeneration and fibrotic repair by producing various extracellular matrix components [38,39,65]. Therefore, the co-expression of Ccl6 and Ccr2 in Arg1+macrophages suggests that these molecules may play a pivotal role in the injury repair process.

Using an AKI mouse model, we further observed that on day 7 post-AKI, cells co-expressing Ccl6, Ccr2, and Arg1 were present in the kidney, whereas these cells were absent in the sham control group. These findings were consistent with our single-cell analysis. Given that Ccr2 and Arg1 are primarily expressed in macrophages, we hypothesize that these Ccl6+Ccr2+Arg1+ cells likely originate from macrophage subpopulations. Interestingly, some Ccl6+Ccr2+ cells were Arg1-negative in the AKI kidneys, suggesting that the M2 polarization process may not yet be fully activated. Future studies will explore the co-expression patterns of Ccl6, Ccr2, and Arg1 at later time points post-AKI. By administering a Ccr2 antagonist via intraperitoneal injection in AKI mice, we found that inhibition of Ccr2 significantly reduced the infiltration of Arg1+F4/80+ cells. Histological analysis of the kidneys on day 7 post-AKI revealed the initiation of fibrotic repair, and inhibition of Ccr2 in vivo attenuated the progression of renal fibrosis. In vitro experiments further demonstrated that Ccl6 promoted the migration of BMDMs and upregulated Arg1 expression, without enhancing their proliferative activity. Taken together, our functional studies preliminarily elucidate that the interaction between macrophage derived Ccl6 and Ccr2 facilitates M2 polarization and induces macrophage migration, which may in turn contribute to the progression of renal fibrosis following AKI.

Although we have identified a potential association between Ccl6+Ccr2+Arg1+macrophages and renal interstitial fibrosis following AKI, our study has certain limitations. First, we did not establish Ccl6 or Ccr2 knockout mouse models. As a result, we were unable to assess the impact of Ccl6/Ccr2 deletion on renal function and fibrosis progression in AKI mice. Second, while we observed a correlation between Ccl6+Ccr2+Arg1+macrophages and the severity of renal fibrosis, and confirmed that Ccl6 promotes the migration and M2 polarization of BMDMs, we did not investigate the specific downstream signaling pathways activated by the Ccl6-Ccr2 interaction that promote fibrosis progression. Previous studies have shown that Ccr2 activation can exacerbate myocardial ischemia-reperfusion injury by promoting the upregulation of the NLRP3/caspase-1/IL-1β inflammatory pathway in macrophages, and it can also enhance cardiac fibrosis by promoting

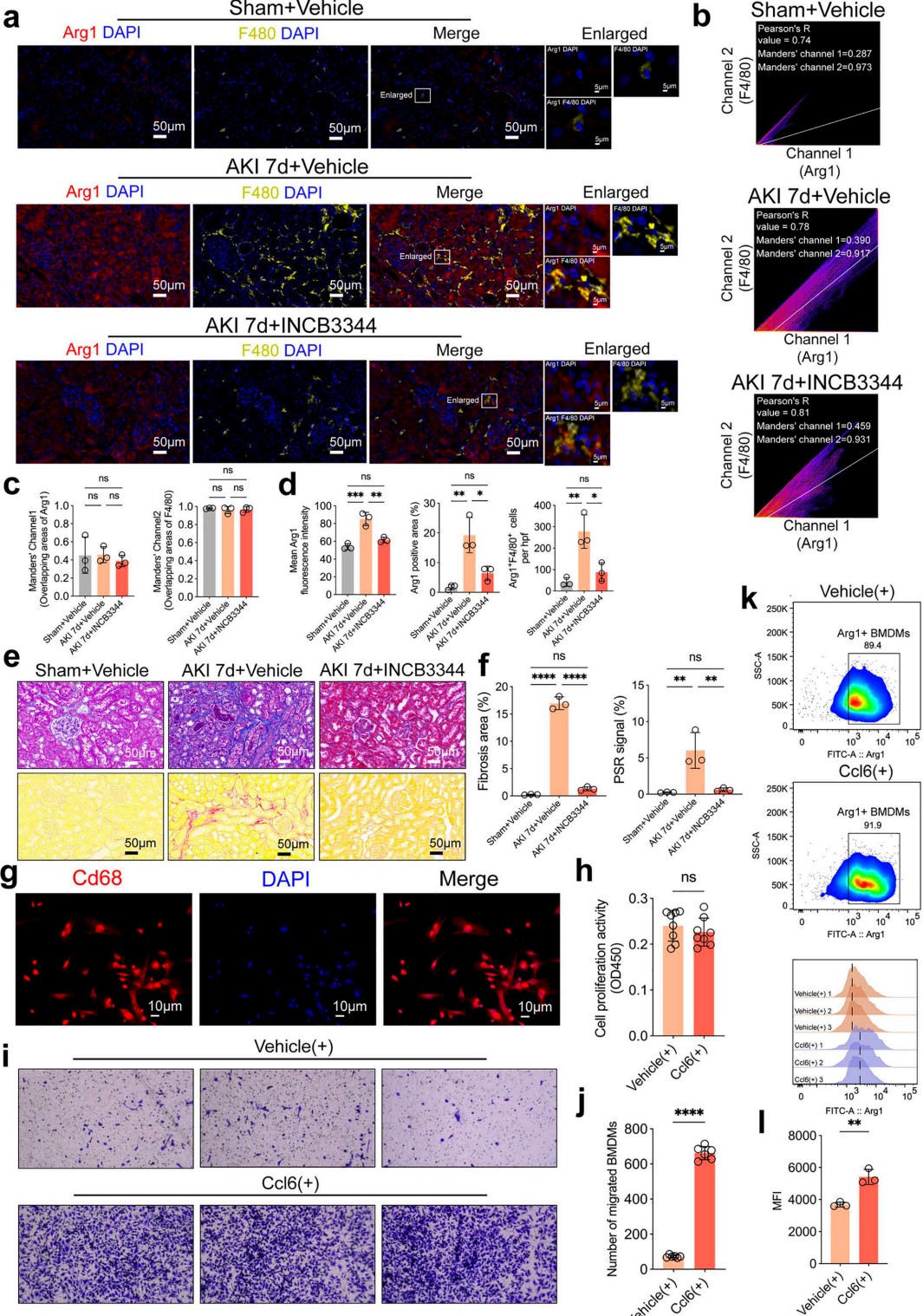

**Fig 6. The role of the Ccl6/Ccr2 axis in macrophage migration, M2 polarization, and renal fibrosis following AKI. (a)** Co-immunofluorescent staining for Arg1 and F4/80 in kidney tissues 7 days post-AKI showed a significant increase in expression compared to the Sham group, which was reduced following Ccr2 antagonist treatment (n = 3). Scale bar: 50 μm. **(b, c)** Co-localization analysis of Arg1 and F4/80 signals revealed a significant correlation,

with no significant differences in Pearson's correlation coefficient or Manders' overlap coefficient among groups. **(d)** Quantification of Arg1 mean fluorescence intensity, Arg1-positive area percentage, and Arg1+F4/80+ double-positive macrophages showed significant increases in the AKI group, which were reduced following Ccr2 blockade. **(e, f)** Masson and Sirius Red staining demonstrated a marked reduction in renal fibrosis upon Ccr2 inhibition (n = 3). Scale bar: 50 μm. **(g)** Confirmation of macrophage identity by Cd68 immunofluorescence staining in isolated bone BMDMs. Scale bar: 10 μm **(h)** CCK8 assay revealed no significant effect of 24-hour Ccl6 treatment on BMDM proliferation (n = 6). **(i, j)** Transwell migration assays showed a significant increase in BMDMs migration in the Ccl6-treated group (n = 6). **(k, l)** Flow cytometry analysis indicated an elevated mean fluorescence intensity of Arg1, confirming that Ccl6 promoted M2 polarization of BMDMs (n = 3). Results are presented as mean ± SD. *$P < 0.05$, **$P < 0.01$, ***$P < 0.001$, *ns*: no significance.

M2 macrophages to release TGF-β [66]. During kidney injury, Ccr2 participates in inducing the migration of monocytes to the site of damage, releasing TGF-β and IL-6, and promoting the progression of renal fibrosis [67]. Ccr2+ monocytes can also promote colonic fibrosis by producing TIMP [68]. Therefore, the downstream signaling pathways activated by the interaction between Ccl6 secreted by macrophages and its receptor Ccr2 on the cell surface in AKI may require further molecular experiments for confirmation. Finally, our analysis was based on publicly available scRNA-seq data from a single time point (day 7 post-uIRI). Although we employed data integration methods to mitigate batch effects, relying solely on an external dataset from a single time window may limit generalizability. Future studies incorporating later-stage AKI datasets and experimental validation at the cellular and molecular levels are needed to confirm whether the co-expression of Ccl6 and Ccr2 directly contributes to renal fibrosis progression. Recent studies have reported that 11β-HSD1 promotes the activation of myofibroblasts, increases collagen deposition, and contributes to the formation of skin scars [69]. Mitochondrial fission and oxidative stress are involved in the epithelial-to-mesenchymal transition (EMT), which accelerates the progression of pulmonary fibrosis [24], Therefore, targeting the regulation of myofibroblast activation and maintaining mitochondrial homeostasis may also play a role in alleviating the progression of fibrosis.

In summary, by analyzing publicly available single-cell data from mice at day 7 post-AKI, we identified the co-expression of Ccl6, Ccr2, and Arg1 in macrophages, with their expression being coordinately upregulated in a time-dependent manner during differentiation. Using the uIRI-induced AKI model, we observed a significant increase in the infiltration of Ccl6+Ccr2+Arg1+macrophages within the OSOM region, which was positively correlated with the degree of interstitial fibrosis. These findings suggest that Ccl6+Ccr2+Arg1+macrophages may represent a distinct subpopulation that promotes renal interstitial fibrosis following AKI. Although targeting the Ccl6/Ccr2 axis holds promise as a therapeutic approach to prevent the progression from AKI to CKD, its efficacy remains to be confirmed through future functional investigations.

## Supporting information

**S1 Fig. Flowchart of this study.**
(PDF)

**S2 Fig. Histological analysis of kidneys from AKI and Sham controls.**
(PDF)

**S3 Fig. Multiplex immunofluorescence reveals the co-localization of Ccl6, Ccr2, and Arg1 in macrophages within kidneys 7 days post-AKI.**
(PDF)

**S4 Fig. Western blot analysis of Ccr2 expression in AKI kidneys.**
(PDF)

**S5 Fig. Ccl6/Ccr2 axis mediates macrophage recruitment, M2 polarization, and fibrosis After AKI.**
(PDF)

**S1. Checklist.**
(PDF)

## Acknowledgments

The authors would like to express their gratitude to the Medical Research Center of Beijing Chaoyang Hospital, Capital Medical University, and the Peking University Applied Lithotripsy Institute for providing the experimental platforms.

## Author contributions

**Conceptualization:** Xin Zheng, Jiayu Wang, Liang Chen.

**Data curation:** Xin Zheng, Jiayu Wang, Liang Chen.

**Formal analysis:** Xin Zheng, Jiayu Wang.

**Funding acquisition:** Liang Chen.

**Investigation:** Xin Zheng, Jiayu Wang, Weinan Chen, Qingquan Xu, Tao Xu.

**Methodology:** Xin Zheng, Jiayu Wang, Weinan Chen, Qingquan Xu, Tao Xu, Liang Chen.

**Project administration:** Xin Zheng, Jiayu Wang, Liang Chen.

**Resources:** Xin Zheng, Liang Chen.

**Software:** Xin Zheng, Jiayu Wang.

**Supervision:** Xin Zheng, Jiayu Wang, Liang Chen.

**Validation:** Xin Zheng, Jiayu Wang, Weinan Chen, Qingquan Xu, Tao Xu, Liang Chen.

**Visualization:** Xin Zheng, Jiayu Wang.

**Writing – original draft:** Xin Zheng, Jiayu Wang.

**Writing – review & editing:** Xin Zheng, Jiayu Wang, Weinan Chen, Qingquan Xu, Tao Xu, Liang Chen.

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
