## [Decision Letter · Decision Letter 0]

13 Jun 2025

Dear Dr. Chen,

Thank you for submitting your manuscript to PLOS ONE. After careful consideration, we feel that it has merit but does not fully meet PLOS ONE’s publication criteria as it currently stands. Therefore, we invite you to submit a revised version of the manuscript that addresses the points raised during the review process.

**ACADEMIC EDITOR:** Thank you for submitting your manuscript to the Journal and as voucan see that the reviewer think your manuscript is interesting and provide valuable comments for your reference. Please submit the revised manuscript ASAP and also include a rebuttal that would clearly list all the responses to the reviewer's comments.

We look forward to receiving your revised manuscript.

Kind regards,

Zhiwen Luo

Academic Editor

PLOS ONE

Journal Requirements:

4. Please include captions for your Supporting Information files at the end of your manuscript, and update any in-text citations to match accordingly. Please see our Supporting Information guidelines for more information: http://journals.plos.org/plosone/s/supporting-information .

Reviewers' comments:

Reviewer's Responses to Questions

**Comments to the Author**

1. Is the manuscript technically sound, and do the data support the conclusions?

Reviewer #1: Yes

Reviewer #2: Yes

2. Has the statistical analysis been performed appropriately and rigorously?

Reviewer #1: Yes

Reviewer #2: Yes

3. Have the authors made all data underlying the findings in their manuscript fully available?

Reviewer #1: Yes

Reviewer #2: Yes

4. Is the manuscript presented in an intelligible fashion and written in standard English?

Reviewer #1: Yes

Reviewer #2: Yes

Reviewer #1: The authors provided a well written piece of work, with conclusions supported by the data shown. The authors are realist about the conclusions that can be drawn from the experiments that were performed, as well as about the limitations of the work. All data was provided, including raw western blot data.

Minor notes:

- Endogenous controls for both qPCR and western blot are not mentioned in the methods section (although WB's control is shown). I would kindly request that the authors add them to the methods section

- I believe that the collection of results clearly indicate the relevance of Ccl6+ Ccr2+ cells that are most likely macrophages, but: one extra staining of CD11b or F4/80 would perfectly drive home the argument of the paper. If it were possible to be added, the paper would greatly benefit from it; otherwise, it would not detract from an otherwise good work.

Reviewer #2: This manuscript presents an interesting and timely investigation into the role of a specific macrophage subpopulation in the transition from acute kidney injury (AKI) to fibrosis. The authors combine an integrated analysis of public single-cell RNA sequencing datasets with in vivo validation in a murine unilateral ischemia-reperfusion injury (uIRI) model. The identification of a Ccl6+Ccr2+Arg1+ macrophage population that correlates with the degree of interstitial fibrosis is a novel finding and points to a potentially important cellular player in this pathological process. The manuscript is well-structured and the data are generally presented clearly. However, the study's conclusions are largely based on correlational evidence, and a significant number of points require clarification and further experimental support to substantiate the claims regarding the functional role of the Ccl6/Ccr2 axis in driving fibrosis.

Major revisions are required:

The central claim is that the Ccl6/Ccr2 axis is associated with, and may promote, renal fibrosis. However, the evidence provided is correlational, demonstrating an association between the number of Ccl6+Ccr2+Arg1+ cells and the extent of fibrosis. To move from correlation to causation and truly test the functional significance of this axis, have you considered functional experiments? For instance, in vivo administration of a Ccr2 antagonist or a neutralizing antibody for Ccl6 in the uIRI model would be required to assess whether blocking this interaction attenuates M2 macrophage infiltration and subsequent fibrosis.

Similarly, can the proposed mechanism be validated in vitro? For example, could bone marrow-derived macrophages (BMDMs) be treated with recombinant Ccl6 protein to assess its effect on macrophage migration, proliferation, or polarization towards an Arg1+ phenotype? This would provide direct evidence for the signaling function of Ccl6 on macrophages.

The single-cell analysis integrates multiple public datasets. While Harmony was used for batch correction, the robustness of the key findings could be strengthened. Could you please show whether the core discoveries—specifically the upregulation of the CCL signaling pathway and the Ccl6/Ccr2 ligand-receptor pair—are consistently observed across the individual AKI datasets, or if the signal is disproportionately driven by one or two of the datasets?

The CellChat analysis identifies macrophages as both the primary source of Ccl6 and a key target via Ccr2. The manuscript suggests this drives macrophage migration. Could you clarify whether this interaction is interpreted as being predominantly autocrine (macrophages stimulating their own migration) or if there is also evidence of a paracrine loop where resident macrophages recruit circulating Ccr2+ monocytes? The distinction is mechanistically important.

The study identifies the Ccl6+Ccr2+Arg1+ population. Using your scRNA-seq data, could you provide a more comprehensive molecular signature for this specific triple-positive subset? What other pro-fibrotic genes (e.g., Spp1, Lgals3, Tgfb1) or matrix-remodeling enzymes are co-expressed in these cells compared to Arg1+ cells that are negative for Ccl6/Ccr2? This would offer a more granular view of their potential function.

The references cited in this article are not sufficient, and there is a lack of in-depth comparative discussion. Background and methodology also require further literature support. Some related research should be cited:

10.15212/bioi-2024-0052

10.15212/bioi-2024-0037

10.15212/bioi-2024-0062

10.1093/burnst/tkac059

10.1093/burnst/tkac052

10.15212/CVIA.2024.0051

10.34133/research.0063

10.5847/wjem.j.1920-8642.2023.057

In the multiplex immunofluorescence analysis (Figure 5d), the claim of identifying triple-positive cells relies on visual assessment of merged images. To make this conclusion more robust, please provide higher magnification images and, critically, quantitative colocalization analysis (e.g., using Pearson's correlation coefficient or Manders' overlap coefficient) to statistically validate the co-expression of all three markers within single cells.

The manuscript structure is logical, but the Discussion section could be strengthened. While the limitations are appropriately acknowledged, the discussion would benefit from a deeper exploration of the potential downstream pathways activated by Ccl6/Ccr2 signaling in macrophages that could lead to a pro-fibrotic phenotype.

Please review the manuscript for professional language and consistency. For example, the heading "Ccl6 and Ccr2 interaction induced macrophage autologous migration" uses the term "autologous migration," which is non-standard. "Autocrine-mediated migration" or rephrasing to "The Ccl6-Ccr2 interaction promotes macrophage migration" would be more appropriate. Please ensure consistent use of terminology throughout.

The conclusions drawn in the Abstract and Conclusion section should be carefully worded to accurately reflect the data. As the study is currently correlational, phrases like "Targeting the Ccl6/Ccr2 axis may attenuate fibrotic progression", while plausible, should be presented as a hypothesis to be tested in future functional studies, rather than a direct conclusion from the present work.

**Do you want your identity to be public for this peer review?** For information about this choice, including consent withdrawal, please see our Privacy Policy

Reviewer #1: No

Reviewer #2: No

---

## [Author Response · Author response to Decision Letter 1]

25 Jul 2025

Response to Reviewer #1 (minor revision)

Reviewer comment 1:

Endogenous controls for both qPCR and western blot are not mentioned in the methods section (although WB's control is shown). I would kindly request that the authors add them to the methods section.

Author's response:

We sincerely thank the reviewer for their valuable feedback and for taking the time to evaluate our manuscript. Regarding the comment on endogenous controls for both qPCR and Western blot, we appreciate the reviewer pointing this out. As per the reviewer’s suggestion, we have now added that GAPDH was used as the endogenous control for both qPCR and Western blot experiments. These details are now included in the Methods section to enhance the clarity and transparency of the experimental procedures. Thank you again for this insightful comment, which has undoubtedly improved the quality of our manuscript.

Reviewer comment 2:

I believe that the collection of results clearly indicate the relevance of Ccl6+ Ccr2+ cells that are most likely macrophages, but: one extra staining of CD11b or F4/80 would perfectly drive home the argument of the paper. If it were possible to be added, the paper would greatly benefit from it; otherwise, it would not detract from an otherwise good work.

Author's response:

We sincerely thank the reviewer for their valuable feedback. In response to your suggestion, as well as the comments from another reviewer, we have conducted an additional experiment to further explore the macrophage identity of Ccl6+Ccr2+Arg1+cells. Specifically, we administered a Ccr2 antagonist (INCB3344) via intraperitoneal injection to uIRI mice, and kidney specimens were collected after 7 days to assess its effect on macrophage polarization in the kidney. We performed fluorescence co-staining for Arg1 and F4/80, which is now included in Figure 6. The results from this additional experiment further support the relevance of Ccl6+ Ccr2+Arg1+cells as macrophages and enhance the manuscript. Thank you again for this helpful suggestion, which has contributed to strengthening the manuscript.

Response to Reviewer #2 (major revision)

1. The central claim is that the Ccl6/Ccr2 axis is associated with, and may promote, renal fibrosis. However, the evidence provided is correlational, demonstrating an association between the number of Ccl6+Ccr2+Arg1+ cells and the extent of fibrosis. To move from correlation to causation and truly test the functional significance of this axis, have you considered functional experiments? For instance, in vivo administration of a Ccr2 antagonist or a neutralizing antibody for Ccl6 in the uIRI model would be required to assess whether blocking this interaction attenuates M2 macrophage infiltration and subsequent fibrosis.

Author's response:

We greatly appreciate the reviewer’s thoughtful and constructive comment. To further explore the causal relationship between Ccl6+Ccr2+Arg1+ macrophages and the progression of renal fibrosis, we performed additional functional experiments. Specifically, we used the uIRI model to induce AKI at day 7, administering a daily intraperitoneal injection of the Ccr2 antagonist (INCB3344) (HY-50674, MCE) 30mg/kg/day in mice. We then performed dual immunofluorescence staining for Arg1 and F4/80 in the kidneys to assess the extent of M2 macrophage infiltration. Furthermore, Masson’s trichrome and Sirius red staining were employed to evaluate the degree of fibrosis. The results show that Ccr2 inhibition reduced M2 macrophage infiltration and alleviated interstitial fibrosis in the kidneys. These additional findings are presented in Figure 6. Thank you again for your valuable suggestion. We believe these new experiments further support the role of the Ccl6/Ccr2 axis in renal fibrosis.

2. Similarly, can the proposed mechanism be validated in vitro? For example, could bone marrow-derived macrophages (BMDMs) be treated with recombinant Ccl6 protein to assess its effect on macrophage migration, proliferation, or polarization towards an Arg1+ phenotype? This would provide direct evidence for the signaling function of Ccl6 on macrophages.

Author's response:

We greatly appreciate the reviewer’s valuable suggestion. To validate the proposed mechanism in vitro, we isolated mouse bone marrow-derived macrophages (BMDMs) and treated them with Ccl6 protein (HY-P7143, MCE) at 100 ng/ml. We then conducted several assays to assess the impact of Ccl6 on macrophage function:

• Migration: Transwell assays indicated that Ccl6 significantly increased the migration of BMDMs.

• Proliferation: The CCK8 assay demonstrated that Ccl6 did not promote BMDM proliferation.

• Polarization: Flow cytometry revealed an increased proportion of Arg1+ BMDMs and higher Arg1 fluorescence intensity, suggesting that Ccl6 drives polarization towards an M2 phenotype.

These findings confirm that Ccl6 plays a role in regulating macrophage migration and polarization without affecting proliferation. The results are shown in Figure 6. Thank you again for your thoughtful recommendation. We believe these additional experiments provide further support for the signaling function of Ccl6 in macrophage.

3. The single-cell analysis integrates multiple public datasets. While Harmony was used for batch correction, the robustness of the key findings could be strengthened. Could you please show whether the core discoveries—specifically the upregulation of the CCL signaling pathway and the Ccl6/Ccr2 ligand-receptor pair—are consistently observed across the individual AKI datasets, or if the signal is disproportionately driven by one or two of the datasets?

Author's response:

We sincerely thank the reviewer for this insightful and constructive comment. As suggested, we performed individual CellChat analyses on each of the AKI datasets to evaluate the robustness of our key findings—specifically the activation of the CCL signaling pathway and the enhancement of the Ccl6–Ccr2 ligand–receptor interaction.

Specifically, compared to controls, the CCL signaling pathway was consistently activated in each individual AKI dataset, and the critical Ccl6–Ccr2 ligand–receptor pair demonstrated strong interactions across all AKI datasets analyzed. These consistent findings suggest that our key discoveries are not disproportionately driven by one or two datasets, but rather reflect shared signaling pathways and ligand–receptor interactions across diverse AKI single-cell datasets. We have added supporting figures to the revised manuscript (see Fig. 2f) to clearly demonstrate this consistency. We hope this additional validation strengthens the confidence in our conclusions, and we greatly appreciate your insightful suggestion.

4. The CellChat analysis identifies macrophages as both the primary source of Ccl6 and a key target via Ccr2. The manuscript suggests this drives macrophage migration. Could you clarify whether this interaction is interpreted as being predominantly autocrine (macrophages stimulating their own migration) or if there is also evidence of a paracrine loop where resident macrophages recruit circulating Ccr2+ monocytes? The distinction is mechanistically important.

Author's response:

We thank the reviewer for this insightful mechanistic question. To address it, we analyzed the expression of Ccl6 and Ccr2 across macrophage clusters using the single-cell FeaturePlot visualization. The annotated FeaturePlot has been incorporated into Fig. 3f to better illustrate the cluster-specific expression patterns of Ccl6 and Ccr2.

Specifically, in Cluster 0, the majority of macrophages exhibit high Ccr2 and low Ccl6 expression, indicating that they are likely responding macrophages, such as recruited or activated macrophages. Notably, we also identified a small subset of cells within Cluster 0 that co-express both Ccl6 and Ccr2 at high levels. This subset suggests the presence of a potential autocrine signaling loop, where macrophages may respond to their own secreted Ccl6. In contrast, Cluster 3 is composed almost entirely of macrophages with high Ccl6 and low Ccr2 expression, consistent with a paracrine model where these cells act as the primary source of the Ccl6 ligand to recruit Ccr2+ macrophages.

Taken together, these data support a model in which the Ccl6/Ccr2 axis functions predominantly through paracrine signaling originating from Cluster 0 and Cluster 3, while a minor autocrine component may exist within Cluster 0 itself. We have clarified this mechanistic interpretation in the revised Results section under the subheading “Trajectory analysis revealed temporal expression consistency of Ccl6-Ccr2 in macrophages” (Page 10).

5. The study identifies the Ccl6+Ccr2+Arg1+ population. Using your scRNA-seq data, could you provide a more comprehensive molecular signature for this specific triple-positive subset? What other pro-fibrotic genes (e.g., Spp1, Lgals3, Tgfb1) or matrix-remodeling enzymes are co-expressed in these cells compared to Arg1+ cells that are negative for Ccl6/Ccr2? This would offer a more granular view of their potential function.

Author's response:

We appreciate the reviewer’s thoughtful suggestion. In our dataset, the Ccl6+Ccr2+Arg1+ macrophage population was exclusively identified within Cluster 3. Notably, we did not observe any Arg1+ macrophage subclusters that lacked expression of either Ccl6 or Ccr2.

Following your advice, we further characterized this triple-positive subset by examining the expression of additional pro-fibrotic genes, including Spp1, Lgals3, and Tgfb1, using FeaturePlot visualization. We found that Spp1 and Lgals3 were highly expressed in the Ccl6⁺Ccr2⁺Arg1⁺ macrophages, while Tgfb1 showed low expression levels. To further explore the matrix-remodeling potential of this subset, we analyzed five representative remodeling enzymes: Mmp9, Mmp12, Timp1, Ctsk, and Adamts4. Among these, only Mmp12 was highly expressed in the Ccl6+Ccr2+Arg1+ population, suggesting that these macrophages may contribute to the progression of renal fibrosis through tissue remodeling mechanisms.

These additional analyses and their visualization have been included in Fig. 4i, and corresponding descriptions have been added to the revised Results section under the heading “The Ccl6-Ccr2 interaction promotes macrophage migration and M2 polarization.”

We thank the reviewer again for this valuable suggestion, which has helped strengthen our functional interpretation of the Ccl6+Ccr2+Arg1+ macrophage subset.

6. The references cited in this article are not sufficient, and there is a lack of in-depth comparative discussion. Background and methodology also require further literature support. Some related research should be cited:

10.15212/bioi-2024-0052

10.15212/bioi-2024-0037

10.15212/bioi-2024-0062

10.1093/burnst/tkac059

10.1093/burnst/tkac052

10.15212/CVIA.2024.0051

10.34133/research.0063

10.5847/wjem.j.1920-8642.2023.057

Author's response:

Thank you for your valuable feedback. In response to your suggestion, I have made improvements in the manuscript. I have included a more in-depth comparative discussion in both the Background and Discussion sections. Additionally, I have incorporated the references you mentioned into the Background, Materials and Methods, and Discussion sections to strengthen the literature support. Furthermore, I have cited all the references you provided, and in the marked-up version of the revised manuscript, these references have been highlighted in bold and red for easy identification. Thank you again for your constructive comments.

7. In the multiplex immunofluorescence analysis (Figure 5d), the claim of identifying triple-positive cells relies on visual assessment of merged images. To make this conclusion more robust, please provide higher magnification images and, critically, quantitative colocalization analysis (e.g., using Pearson's correlation coefficient or Manders' overlap coefficient) to statistically validate the co-expression of all three markers within single cells.

Author's response:

Thank you for your constructive suggestion regarding the multiplex immunofluorescence analysis (Figure 5d). As per your recommendation, we have selected higher magnification images to provide a clearer visualization of the triple-positive cells. Furthermore, we performed quantitative colocalization analysis using Fiji (ImageJ) software. Specifically, we calculated both Pearson’s correlation coefficient and Manders’ overlap coefficient to statistically validate the co-expression of Ccl6, Ccr2, and Arg1 within single cells. The results of this supplementary analysis have now been included and are presented in the revised Figure 5. We believe these additions strengthen the robustness of our conclusion.

8. The manuscript structure is logical, but the Discussion section could be strengthened. While the limitations are appropriately acknowledged, the discussion would benefit from a deeper exploration of the potential downstream pathways activated by Ccl6/Ccr2 signaling in macrophages that could lead to a pro-fibrotic phenotype.

Author's response:

Thank you for your valuable feedback on our manuscript. We greatly appreciate your suggestion to strengthen the Discussion section by exploring the potential downstream pathways activated by the Ccl6/Ccr2 signaling in macrophages, which could lead to a pro-fibrotic phenotype. In response, we have revised the manuscript to address this aspect in more detail. We now include a discussion of existing research that highlights how Ccr2 activation can promote inflammation via the NLRP3/caspase-1/IL-1β pathway, leading to increased myocardial injury and fibrosis in heart tissue. We also discuss the role of Ccr2 in kidney injury, where it induces the migration of monocytes to the site of damage and activates Tgf-β and IL-6, which in turn promote renal fibrosis progression. Additionally, we note that CCR2+ monocytes can produce TIMP, facilitating fibrosis in the colon. Therefore, we emphasize that the downstream signaling pathways activated by the interaction of Ccl6 with its receptor Ccr2 in macrophages during AKI require further molecular experiments to be conclusively verified. We hope this addition better aligns with your expectations and further clarifies the potential implications of the Ccl6/Ccr2 axis in fibrosis. Once again, we sincerely appreciate your insightful comments, which have helped us improve the quality of our manuscript.

9. Please review the manuscript for professional language and consistency. For example, the heading "Ccl6 and Ccr2 interaction induced macrophage autologous migration" uses the term "autologous migration," which is non-standard. "Autocrine-mediated migration" or rephrasing to "The Ccl6-Ccr2 interaction promotes macrophage migration" would be more appropriate. Please ensure consistent use of terminology throughout.

Author's response:

Thank you for your valuable feedback. As per your suggestion, I have revised the title to "The Ccl6-Ccr2 interaction promotes macrophage migration and M2 polarization", which more accurately reflects the content. Additionally, I have ensured consistent use of this terminology throughout the manuscript and have avoided the use of the non-standard term "autologous migration".

10. The conclusions drawn in the Abstract and Conclusion section should be carefully worded to accurately reflect the data. As the study is currently correlational, phrases like "Targeting the Ccl6/Ccr2 axis may attenuate fibrotic progression", while plausible, should be presented as a hypothesis to be tested in future functional studies, rather than a direct conclusion from the present work.

Author's response:

Thank you for your insightful comment. Based on your feedback, I have carefully revised the Abstract and Conclusion sections to reflect the correlational nature of our study more accurately. The phrase "Targeting the Ccl6/Ccr2 axis may attenuate fibrotic progression" has been reworded to present it as a hypothesis that requires further validation through future functional studies. I appreciate your thoughtful suggestion, and I believe these revisions enhance the clarity and accuracy of the manuscript.

---

## [Decision Letter · Decision Letter 1]

26 Aug 2025

Single-cell transcriptomic analysis reveals the association of Ccl6+Ccr2+Arg1+ macrophages with renal interstitial fibrosis in AKI

PONE-D-25-18641R1

Dear Dr. Chen,

We’re pleased to inform you that your manuscript has been judged scientifically suitable for publication after peer reviews and will be formally accepted for publication once it meets all outstanding technical requirements.

Within one week, you’ll receive an email detailing the required amendments. When these have been addressed, you’ll receive a formal acceptance letter ,and your manuscript will be scheduled for publication.

Kind regards,

Vipula Rasanga Bataduwaarachchi, MD

Academic Editor

PLOS ONE

Reviewers' comments:

Reviewer's Responses to Questions

**Comments to the Author**

Reviewer #3: All comments have been addressed

Reviewer #4: All comments have been addressed

2. Is the manuscript technically sound, and do the data support the conclusions?

Reviewer #3: Yes

Reviewer #4: Yes

3. Has the statistical analysis been performed appropriately and rigorously?

Reviewer #3: Yes

Reviewer #4: Yes

4. Have the authors made all data underlying the findings in their manuscript fully available?

Reviewer #3: Yes

Reviewer #4: Yes

5. Is the manuscript presented in an intelligible fashion and written in standard English?

Reviewer #3: Yes

Reviewer #4: Yes

Reviewer #3: The authors made significant revisions to the manuscript, and most of the concerns raised by reviewers have been addressed in the revised version. The article is publishable in this form.

Reviewer #4: it looks ok

the paper Single-cell transcriptomic analysis reveals the association of Ccl6+Ccr2+Arg1+

macrophages with renal interstitial fibrosis in AKI and should accept in PLOS ONE

**Do you want your identity to be public for this peer review?** For information about this choice, including consent withdrawal, please see our Privacy Policy

Reviewer #3: No

Reviewer #4: No

---

## [Editor Report · Acceptance letter]

PONE-D-25-18641R1

PLOS ONE

Dear Dr. Chen,

I'm pleased to inform you that your manuscript has been deemed suitable for publication in PLOS ONE. Congratulations! Your manuscript is now being handed over to our production team.

Kind regards,

on behalf of

Dr. Vipula Rasanga Bataduwaarachchi

Academic Editor

PLOS ONE